# Predictors of time to death among preterm neonates admitted to neonatal intensive care units at public hospitals in southern Ethiopia: A cohort study

**Alo Edin Huka**[1]*, **Lemessa Oljira**[2], **Adisu Birhanu Weldesenbet**[2], **Abdulmalik Abdela Bushra**[2], **Ibsa Abdusemed Ahmed**[2], **Abera Kenay Tura**[3,4], **Angefa Ayele Tuluka**[1]

1 Department of Epidemiology, School of Public Health, Institute of Health, Bule Hora University Bule Hora, Ethiopia, 2 School of Public Health, College of Health and Medical Sciences, Haramaya University, Harar, Ethiopia, 3 School of Nursing and Midwifery, Collage of Health and Medical Sciences, Haramaya University, Harar, Ethiopia, 4 Department of Obstetrics and Gynecology, University Medical Centre Groningen, University of Groningen, Groningen, The Netherlands

* jiblosa1@gmail.com

**Data Availability Statement:** All relevant data are within the paper and its Supporting information files.

## Abstract

### Background

Although the survival of preterm neonates has improved, thanks to advanced and specialized neonatal intensive care, it remains the main reason for neonatal admission, death, and risk of lifelong complication. In this study, we assessed time to death and its predictors among preterm neonates admitted to neonatal intensive care units (NICU) at public hospitals in southern Ethiopia.

### Methods

A hospital based retrospective cohort was conducted among preterm neonates admitted to NICU at public hospitals in west Guji and Borena zones, Oromia National Regional State, southern Ethiopia. Simple random sampling technique was used to select records of preterm neonates admitted to both major hospitals in the study area. Data on neonatal condition, obstetric information, and status at discharge were collected from admission to discharge by trained research assistant through review of their medical records. Kaplan Meir curve and Log rank test were used to estimate the survival time and compare survival curves between variables. Cox-Proportional Hazards model was used to identify significant predictors of time to death at p<0.05.

### Result

Of 510 neonates enrolled, 130(25.5%; 95% CI: 22–29) neonates died at discharge or 28days. The median survival time was 18 days with an interquartile range of (IQR = 6, 24). The overall incidence of neonatal mortality was 47.7 (95% CI: 40.2–56.7) per 1000 neonatal days. In the multivariable cox-proportional hazard analysis, lack of antenatal care (AHR: 7.1; 95%CI: 4–12.65), primipara (AHR: 2.3; 95% CI: 1.16–4.43), pregnancy complications

**Funding:** The authors received no specific funding for this work.

**Competing interests:** The authors have declared that no competing interests exist.

**Abbreviations:** AHR, Adjusted Hazard Ratio; ANC, Antenatal Care; APGAR, Appearance pulse grimace activity respiration; CI, Confidence Interval; CS, Caesarean section; DM, Diabetic Mellitus; FMoH, Federal Minster of Health; HIV, Human Immunodeficiency Virus; HR, Hazard Ratio; HSTP, Health Sector Transformation Plan; KMC, Kangaroo Mother Care; nCPAP, Nasal continuous positive air pressure; NICU, Neonatal Intensive Care Unit; PNA, Perinatal Asphyxia; PROM, Pre-labour rupture of membrane; RDS, Respiratory Distress Syndrome; SDGs, Sustainable Development Goals; SVD, Spontaneous vaginal delivery; UNICEF, United Nations international emergency children fund; WHO, World Health Organization.

(AHR: 3.4; 95% CI: 1.94–6.0), resuscitation at birth (AHR: 2.1, 95% CI: 0.28–0.77) and not receiving Kangaroo mother care (AHR: 9.3, 95% CI: 4.36–19.9) were predictors of preterm neonatal death.

## Conclusion

Despite admission to NICU for advanced care and follow up, mortality of preterm neonates was found to be high in the study settings. Addressing major intrapartum complications is required to improve survival of neonates admitted to NICU.

## Introduction

Preterm birth is defined as a live birth before 37 completed weeks of pregnancy [1] and remains one of the common causes of hospitalization, death and long-term complications of the neonatal period. Although survival of preterm neonates has improved significantly, partly thanks to the advanced and specialized care in the neonatal intensive care unit (NICU), it still remains the main reason for neonatal admission, death, and risk of lifelong complication [2]. Globally, the sub-Saharan Africa has the highest newborn mortality rate 27 deaths per 1,000 live births, followed by Central and Southern Asia 25 deaths per 1,000 live births [3]. Similarly, Ethiopia is one of the top five countries where almost 50% of all the global neonatal mortality occurred [3].

The current level of neonatal death reflects that Ethiopia is far behind the goal set by the national Health Sector Transformation Plan (HSTP-I) which was 10 deaths per 1,000 live births by 2020 [4] and the sustainable development goals (SDGs), target of reducing neonatal mortality to 12 deaths per 1,000 live births by 2030 [4, 5]. Neonatal mortality in Ethiopia is still high despite the country's goal of reducing it from 28 to 11 per 1000 by 2020 as part of the national newborn and child survival strategy (2015–2020) [6]. Study on increasing trends of under-five mortality in the aftermath of Millennium Development Goal in Eastern Ethiopia revealed that overall under five mortality rate was 46.3 per 1000 live births with significant increase from 27.9 in 2015 to 54.7 in 2020 [7].

Despite all strategies and interventions that have been implemented to reduce neonatal mortality, especially preterm neonatal death, Ethiopia's neonatal mortality rate is unacceptably high. Although admission to the NICU will definitely improve the survival of preterm neonate, there is paucity of data on survival of preterm neonates and its determinants in many parts of Ethiopia in general, and in this pastoral community in particular.

The aim of this study was to determine time to death and its predictors among preterm neonates admitted to NICU at public hospitals in pastoral community in southern Ethiopia.

## Materials and methods

### Study design and setting

A hospital based retrospective cohort study was conducted among preterm neonates admitted to the NICU of two major public hospitals (Bule Hora and Yabelo) in West Guji and Borana Zones of Oromia, a predominantly pastoral setting. All neonates admitted from September 11, 2018 to September 10, 2021 in both hospitals were included. Bule Hora hospital, located in the capital of Guji Zone, is a zonal hospital with annual delivery of 3,250.

The hospital is a comprehensive general bedded hospital with 22 beds in the NICU. In this hospital neonatology ward accommodate 16 beds in including 8 total beds from NICU ward.

According to 2019/21 health department health management information system report, while Yabelo general hospital has 6,948 total number of birth attended by skilled health personnel from 2018 to 2021. In this hospital obstetrics and gynecological ward accommodate 24 total number of beds. Annual number of neonatal admissions including preterm neonates estimated to be 560. Nurse to patient ratio and physician to patient ratio is 1:2 and 1:12 respectively.

## Population and sampling

All preterm neonates admitted to NICU at public hospitals of West Guji and Borena zones constitute the source population while all preterm neonates admitted in Bule Hora and Yabelo Hospitals NICU from September 11, 2018 to September 10, 2021 were the study population. Neonates with incomplete records: date of admission, date of discharge, vital status at discharge were excluded.

The sample size was calculated by using STATA 14.1 by considering the following assumptions: hazard ratio (HR) (1.62) of the selected covariate (presence of jaundice) and probability of failure (event) 0.288 were taken from a previous study [8]. Assuming probability of type I error or alpha 0.05, power of the study 80% and withdrawal probability of 0.1. Presence of jaundice was selected out of 5 covariates since it provides a maximum sample. Accordingly, the calculated sample size was 522. Simple random sampling technique was used to select a predetermined sample size by using their unique medical registration number (obtained from the admission register) as a sampling frame.

## Data collection

The checklist for this study was adapted from neonates' medical card and logbook with modification based on literature review. The data were extracted from each individual neonate's medical card and logbook using a structured checklist. The checklist consisted of the information on maternal and neonatal socio demographic factors, neonatal related factors, maternal medical and obstetric-related factors.

Since the data were collected from neonates' medical card, the data collectors were trained on how to extract appropriate data and there was close daily monitoring and supervision at the data collection site.

## Data processing and analysis

Data was entered to Epi Data 3.1 and cleaned and analyzed using STATA 15. Kaplan–Meier curve was used to estimate median survival time, cumulative probability of survival, and compare survival difference between the different covariates. Log rank test was used to compare statistical survival difference between categories of different explanatory variables. Life table was used to estimate the cumulative probability of survival at the different time intervals.

Multi-collinearity was checked using variance inflation factor (VIF<10) indicating nonexistence of multicollinearity. In bivariable analysis crude hazard ratio test was carried out to identify candidate variables for the multivariable Cox regression model at p-value < 0.25. Variables with p-value <0.25 was entered into the multivariable Cox regression model to determine the predictors of preterm neonatal death. Proportional hazard assumption was assessed both graphically and Schoenfeld residual global test and PH assumption was met (chi2 = 2.14 Prob>chi2 = 0.0965). Hazard Ratios (HR) with 95% Confidence Intervals (CI) was used to assess the relationship between predictors associated with the occurrence of preterm neonatal death. Statistical significance was declared at p-value < 0.05.

### Ethical considerations

The study protocol was reviewed and approved by the Institutional Health Research Ethics Review Committee (Ref no: IHRERC/193/2021/YYY) of College of Health and Medical Sciences, Haramaya University, Ethiopia. As the study was retrospective, the need for individual informed consent was waived. However, informed consent was obtained from the medical director and head of NICU of both hospitals. Confidentiality of the patient was ensured using anonymous data collection and authors had no access to individual identifier throughout the research process.

## Results

### Socio-demographic factors

Of the 522 neonatal records reviewed, 510 (97.7%) which met the enrollment requirements were included. However, 12 charts were excluded (5 unrecorded dates of admission, 4 unrecorded dates of discharge, 3 of the charts were not available at the time of data collection). Out of 510 neonates 261(51.2%) were from Yabelo general hospital while 249(48.8%) of them were from Bule hora general hospital. Almost a half of neonates 270(52.9%) were male and near to two-thirds of their mothers, 349(68.4%), came from rural area. Majority of the mothers, 353 (69.2%), were between 20–34 years with median age of the 26 years (IQR = 22, 30) (Table 1).

### Obstetrics characteristics and medical conditions

Two third (65.9%) of the women had at least antenatal care. Nearly half (46.9%) of the women were multiparous. More than half (57.1%) gave birth at hospital and (76.9%) was spontaneous vaginal delivery (Table 2).

### Neonates related factors

Six in ten (60.2%) of the neonate had a low birth weight (1500–2499 gm), with a mean weight of 1770 (±615 SD) gm. Four hundred and five neonates (79%) cried immediately after birth and resuscitation was performed for 41.8% of preterm neonates studied. About 136 (26.7%) had a body. More than a half 299(58.6%) of the neonates were diagnosed with neonatal sepsis

**Table 1. Socio-demographic characteristics of mothers of preterm neonates admitted to NICU at Bule Hora and Yabelo general hospitals from September 11, 2018 to September 10, 2021(n = 510).**

| Characteristics | Frequency | Percentage |
|---|---|---|
| **Hospital** | | |
| Yabelo | 261 | 51.2 |
| Bule hora | 249 | 48.8 |
| **Age of mother** | | |
| <20 | 103 | 20.2 |
| 20–34 | 353 | 69.2 |
| ≥35 | 54 | 10.6 |
| **Maternal residence** | | |
| Urban | 161 | 31.6 |
| Rural | 349 | 68.4 |
| **Sex of neonates** | | |
| Male | 270 | 52.9 |
| Female | 240 | 47.1 |

**Table 2. Obstetric characteristics of mothers of preterm neonates admitted to NICU at Bule Hora and Yabelo general hospitals from September 11, 2018 to September 10, 2021(n = 510).**

| Characteristics | Frequency | Percentage |
|---|---|---|
| **Antenatal care follow up** | | |
| Yes | 336 | 65.9 |
| No | 174 | 34.1 |
| **Level of ANC follow up(n = 336)** | | |
| 1–3 visits | 272 | 81 |
| ≥4 visits | 64 | 19 |
| **Parity** | | |
| Primipara | 119 | 23.3 |
| Multipara | 239 | 46.9 |
| Grand multipara | 152 | 29.8 |
| **Complication during last pregnancy** | | |
| Yes | 132 | 25.9 |
| No | 378 | 74.1 |
| **Previous bad obstetrics history** | | |
| Yes | 87 | 17.1 |
| No | 423 | 82.9 |
| **Type of pregnancy** | | |
| Singleton | 412 | 80.8 |
| Multiple | 98 | 19.2 |
| **Onset of labor** | | |
| Elective caesarean section | 80 | 15.7 |
| Spontaneous | 375 | 73.5 |
| Induced | 55 | 10.8 |
| **Place of birth** | | |
| Home | 92 | 18 |
| Health center | 127 | 24.9 |
| Hospital | 291 | 57.1 |
| **Mode of deliver** | | |
| Spontaneous vaginal delivery | 392 | 76.9 |
| Caesarean section | 115 | 22.5 |
| Instrument assisted delivery | 13 | 0.6 |
| **Duration of labor in hours** | | |
| < 4hrs | 47 | 11.1 |
| 4-18hrs | 322 | 76.3 |
| >18hrs | 53 | 12.6 |
| **Preeclampsia** | | |
| Yes | 107 | 21 |
| No | 403 | 79 |
| **Diabetic Mellitus** | | |
| Yes | 17 | 3.3 |
| No | 493 | 96.7 |
| **Anemia** | | |
| Yes | 12 | 2.4 |
| No | 498 | 97.6 |

at the admission. From the total 510 neonates 237(46.5%) had received kangaroo mother care during the hospital stay (Table 3).

## Survival status of preterm neonates

Neonates were followed for a minimum of 1 day and maximum of 28 days with a median follow-up period of 4 days. At the end of follow-up period, 130(25.5%) (95% CI: 22, 29) of the neonates died. The total time at risk for neonates was 2723 neonatal days. The overall incidence of neonatal mortality rate was 47.7 (95% CI: 40.2, 56.7) per 1000 neonatal days. From all deaths, about 58 (45%) of the neonates died within first 24 hours and about 48(37.2%) of the neonatal deaths occurred in the first 1 week of life.

The cumulative probability of survival at the first, seventh and 28th days was 80.1% (95% CI: 76–84), 62% (95%CI: 56–68) and 45.6% (95%CI: 34–56) respectively. The overall median survival time was 18 days with an interquartile range of (IQR = 6.24). Overall median Length of hospital stay for preterm neonates under the study was 4 days, with an interquartile range of (3, 7) neonates' days (Table 4).

The Kaplan Meier failure and log-rank test was used to compare hazard of death between preterm neonates with the different conditions The Kaplan Meier failure curve shows the probabilities of failure at 10, 20, and 28 days of follow up time which indicates increasing trend over time (Fig 1).

The log rank test revealed that the survival trend or period to neonatal mortality differed significantly between the categories of neonatal mortality: kangaroo mother care (Log rank test, $x^2$ = 102.98), neonatal sepsis (Log rank test, $x^2$ = 8.54), ANC (Log rank test, $x^2$ = 167.2) and complication during last pregnancy (Log rank test, $x^2$ = 106.23) (Fig 2).

## Predictors of time to death

In the bi-variable cox regression analysis, maternal residence, antenatal care follow up, complication during last pregnancy, previous bad obstetrics history, place of birth, mode of delivery, preeclampsia, diabetic mellitus, gestational age, weight for gestational age, neonate cried immediately at birth, bag and mask resuscitation at birth, perinatal asphyxia at birth, neonate diagnosed with respiratory distress, hypoglycemia, neonatal sepsis, phototherapy, nasal continuous positive airway pressure, not receiving kangaroo mother care and radiant warmer were predictors of mortality at p<0.05. However, the multivariable cox-proportional hazard results revealed that the hazard of death among neonates born to mothers who had no antenatal care was 7 times higher compared to those neonates born to mothers who had at least one antenatal care during index pregnancy (AHR: 7.1, 95%CI: 4–12.65). There was a 2-fold increase in the hazard of mortality among preterm infants born to primipara compared to those born to multipara mothers (AHR: 2.3, 95%CI: 1.16–4.43).

This study revealed that babies from mother with pregnancy complication were more likely to die compared to those born to a mother with no pregnancy complication (AHR: 3.4, 95% CI: 1.94–6). The hazard of death among neonates who were resuscitated at birth was twice more likely than those who were not resuscitated (AHR: 2.1, 95%CI: 0.28–0.77). Finally, there was a 9-fold increase in the hazard of mortality among neonates who did not receive kangaroo mother care compared to their counterparts (AHR: 9.3, 95%CI: 4.36–19.9) (Table 5).

## Discussion

This study was conducted to determine time to death and its predictors among preterm neonates admitted to the neonatal intensive care unit in two general hospitals in a predominantly pastoral community in southern Ethiopia. We found that the overall incidence of neonatal

**Table 3. Characteristics of preterm neonates admitted to NICU at Bule Hora and Yabelo general hospitals from September 11, 2018 to September 10, 2021(n = 510).**

| Characteristics | Frequency | Percentage |
|---|---|---|
| Gestational age | | |
| Extremely preterm | 39 | 7.6 |
| very preterm | 79 | 15.5 |
| moderate preterm | 138 | 27.1 |
| late preterm | 254 | 49.8 |
| Weight for gestational age at birth | | |
| Small | 252 | 49.4 |
| Appropriate | 258 | 50.6 |
| Weight of neonate (gm) | | |
| <1000 | 26 | 5.1 |
| 1000–1499 | 133 | 26.1 |
| 1500–2499 | 307 | 60.2 |
| ≥2500 | 44 | 8.6 |
| Fifth minute APGAR score | | |
| <7 | 346 | 74.1 |
| ≥7 | 121 | 25.9 |
| Newborn cry immediately at birth | | |
| Yes | 405 | 79.4 |
| No | 105 | 20.6 |
| Bag and mask resuscitation at birth | | |
| Yes | 213 | 41.8 |
| No | 297 | 58.2 |
| Newborns temperature within 1 h of admission | | |
| ≤32 | 46 | 9 |
| 32.1–34 | 113 | 22.2 |
| 34.1–35 | 117 | 23 |
| 35.1–36 | 136 | 26.7 |
| ≥36 | 97 | 19.1 |
| Peri-natal asphyxia diagnosed at birth | | |
| Yes | 83 | 16.3 |
| No | 427 | 83.7 |
| Newborn diagnosed with respiratory distress | | |
| Yes | 116 | 22.7 |
| No | 394 | 77.3 |
| Hypothermia diagnosed at admission | | |
| Yes | 258 | 50.6 |
| No | 252 | 49.4 |
| Hypoglycemia diagnosed at admission | | |
| Yes | 97 | 19 |
| No | 413 | 81 |
| Jaundice | | |
| Yes | 21 | 4.1 |
| No | 489 | 95.9 |
| Newborn diagnosed with sepsis | | |
| Yes | 299 | 58.6 |
| No | 211 | 41.4 |

(*Continued*)

**Table 3.** (Continued)

| Characteristics | Frequency | Percentage |
|---|---|---|
| Neonate received photo therapy | | |
| Yes | 44 | 8.6 |
| No | 466 | 91.4 |
| Neonate received continuous positive airway pressure (nCPAP) | | |
| Yes | 143 | 28 |
| No | 367 | 72 |
| Newborn received kangaroo mother care | | |
| Yes | 237 | 46.5 |
| No | 273 | 53.5 |
| Newborn heated with radiant warmer | | |
| Yes | 308 | 60.4 |
| No | 202 | 39.6 |

mortality rate was 47.7 (95% CI: 40.2, 56.7) per 1000 neonatal days with median survival time of 18 days. Predictors of death were no antenatal care, birth to primipara mother, obstetric complications, resuscitation at birth, and no kangaroo mother care.

Our finding is consistent with finding from Pakistan (47.3 per 1000 neonatal days) [9]. But higher than Tanzania (6.5 per 1,000) [10], and other parts of Ethiopia :- Wolaita Sodo (27 per 1000 person- days) [11], in Jimma (28.9 per 1, 000 neonate-days) [12], Debre Markos (29.2 per1000 person-day) [13], and Tikur Anbessa (39.1 per 1000-person day) [14] and it is lower than study conducted in Mizan Tepi (62deaths per 1000 person-days) [15]. This might be related with level of care differences. For example, settings with specialized and well-equipped neonatal care facilities could provide better care to the preterm neonates compared to the resource limited health care settings [16] and Similarly, socio demographic variation across the study areas might explain some of the difference. Given our study is in a predominantly pastoral community, women will seek care only in case of severe conditions.

The proportion of preterm neonatal death at the end of follow up period was (25.5% (95% CI: 22, 29). This is consistent with the findings from the studies conducted in Gondar (25.2%) [17], teaching hospitals in Addis Ababa (25.3%) [18], and in Jimma University Medical Center (25.1%) [12].

It is however, higher than findings from Uganda (8%) [19], in Woliata (16.5%) [11], and systematic review and meta-analysis in Ethiopia (19.2%) [20]. proportion of preterm neonatal death is lower than the studies reported from Mizan Tepi (35%) [15], Jimma University specialized hospital (34.9%) [21], tertiary hospital of Uganda (31.6%) [22], Addis Ababa

**Table 4. Failure probability of preterm neonates admitted to NICU at Bule Hora and Yabelo general hospitals from September 11, 2018 to September 10, 2021 (n = 510).**

| Time in days | Total at the beginning | Death | censored | Failure probability% | 95%CI |
|---|---|---|---|---|---|
| 0–5 | 510 | 82 | 196 | 20 | (16.4,24.1) |
| 5–10 | 232 | 38 | 125 | 37.9 | (32.3,44) |
| 10–15 | 69 | 6 | 37 | 45.2 | (38,53.1) |
| 15–20 | 26 | 3 | 16 | 54.4 | (43.5,65.9) |
| 20–25 | 7 | 0 | 6 | 54.4 | (43.5,65.9) |
| 25–28 | 1 | 1 | 0 | 1 | |

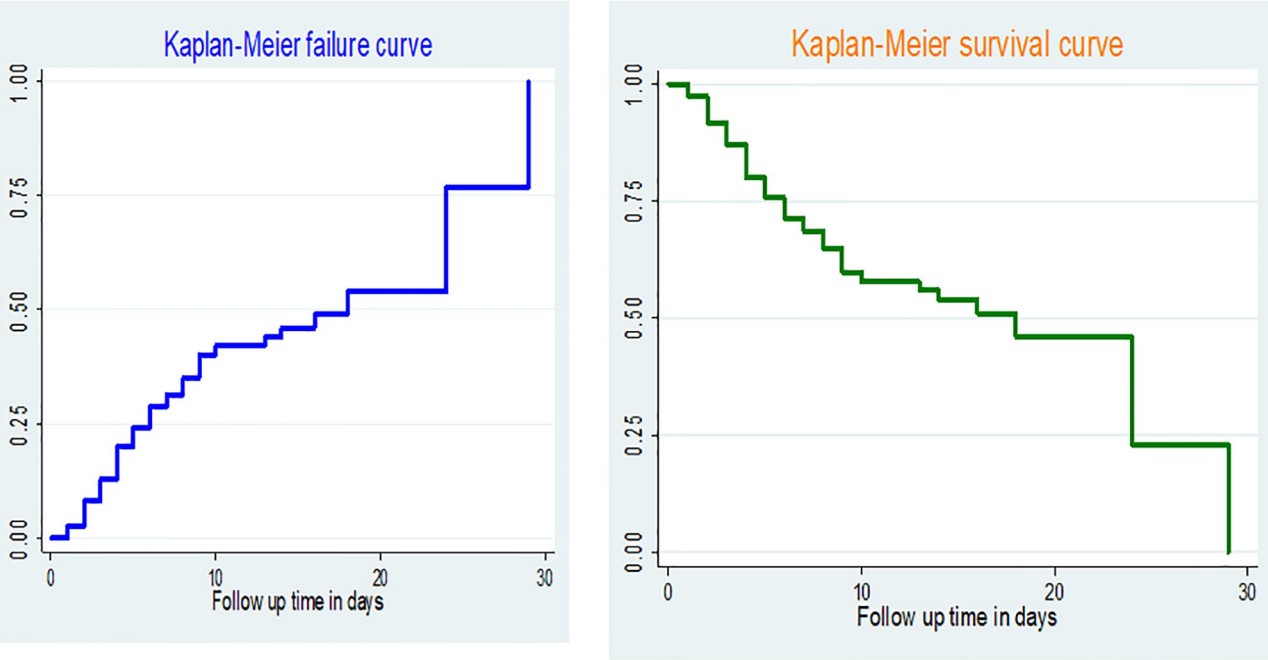

**Fig 1. Kaplan-Meier failure and survival curve for overall time to death among preterm neonates admitted to NICU at Yabelo and Bule Hora hospitals from September 10, 2018 to September 11, 2021 (n = 510).**

University specialized hospital (29.7%) [14], Gondar (28.8%) [8], Debre Markos referral hospital (27.11%) [13], and France (27%) [23]. The variation between these studies could be attributable to differences in the quality of service provided.

We found that the hazard of death among neonates born to mothers who had no antenatal care was 7 times higher compared to those who had at least one antenatal care in index pregnancy. This is supported by the studies conducted in South Western Uganda [16], University of Gondar [17], Bombe primary hospital southern Ethiopia [24], and Mizan Tepi University Teaching hospital [15]. Possible reason could be lack of ANC visits may result in sufficient pregnancy monitoring, which may lead to neonatal problems during and after birth, which may be related with an increased risk of neonatal death.

There was a 2-fold increase in the hazard of mortality among infants born to primipara mothers compared to those born to multipara mothers. this is line with Tigrai [25] and Jimma [26]. This could be linked to the higher risk of unfavorable neonatal outcomes and intrapartum complications among of primipara mothers.

This study revealed that babies from mother with pregnancy complication were more likely to die than those from mother with no pregnancy complication. This is in line with the study from Sudan [27], Jimma Ethiopia [12] and Indonesia [28]. This could be explained by the fact that pregnancy complications impair the fetus's health and can contribute to preterm delivery, which can lead to life-threatening preterm complications, potentially increasing the risk of newborn death. Association of obstetric complications and adverse perinatal outcomes has been reported previously in [29].

Hazard of death among neonates who were resuscitated at birth were twice more likely than those who were not resuscitated at birth. This is in line with the study in Uganda [22] and southern Ethiopia [11]. The possible explanation could be resuscitation is given to the neonates with low 5th Minute APGAR score or failed to cry at birth.

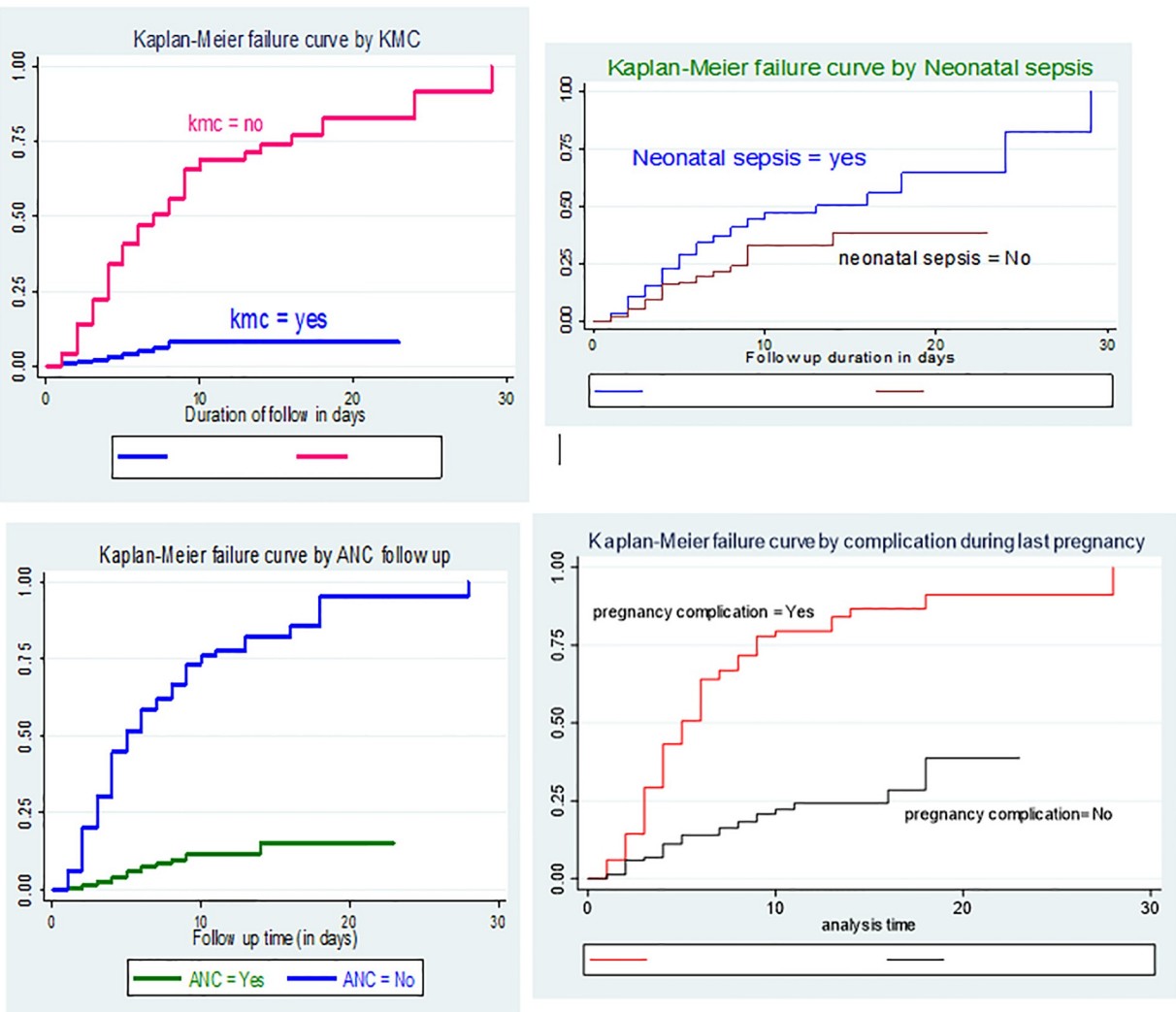

**Fig 2. Kaplan-Meier failure curves of time to death among exposed and unexposed preterm neonates admitted to NICU at Yabelo and Bule Hora hospitals from September 10, 2018 to September 11, 2021 (n = 510).**

This study revealed 9-fold increase in hazard to mortality among preterm neonates who had not received KMC compared to those who were received KMC during hospital stay. This is consistent with the findings from the study in Uganda [16], Mizan Tepi [15], in Jimma [12] and Gondar Ethiopia [8]. There are various mechanisms through which immediate kangaroo mother care could be beneficial. The baby is more likely to be colonized by the mother's protective microbiome and to obtain early breastfeeding because the mother and baby are in close contact from birth. In addition, there is less handling of the newborn by other individuals, lowering the risk of infection.

## Strength and limitation of the study

This study's strength is in its attempt to determine the factors that predict the time to death in preterm neonates admitted to the NICU, highlighting for the professional in the field when to be particularly concerned, especially in a resource-constrained settings. Additionally,

**Table 5. Multivariable cox-proportional hazard regression for predictors of time to death among preterm neonates admitted to NICU at Bule Hora and Yabelo general hospitals from September 11, 2018, to September 10, 2021(n = 510).**

| Variable | Category | Survival status | | CHR 95% CI | AHR 95% CI |
|---|---|---|---|---|---|
| | | Died (N, %) | Censored (N %) | | |
| Age of mother | <20 | 16(3.1) | 87(17.1) | 0.6(0.35,1.01) | 0.6(0.28,1.29) |
| | 20–34 | 97(19) | 256(50.2) | 1 | 1 |
| | ≥35 | 17(3.3) | 37(7.25) | 1.28(0.76,2.15) | 0.77(0.38,1.56) |
| Maternal residence | Urban | 48(9.4) | 113(22.2) | 1 | 1 |
| | Rural | 82(16.1) | 267(52.3) | 0.69(0.48,0.99) * | 0.77(0.47,1.27) |
| ANC | Yes | 20(3.9) | 316(62) | 1 | 1 |
| | No | 110(21.6) | 64(12.6) | 11.34(7.03,18.27)*** | 7.1(4,12.65) *** |
| Parity | Primipara | 24(4.7) | 95(18.6) | 0.73(0.46,1.17) | 2.3(1.16,4.43) * |
| | Multipara | 66(12.9) | 173(3.9) | 1 | 1 |
| | Grand multipara | 40(7.8) | 112(22) | 0.97(0.65, 1.44) | 1.5(0.88,2.42) |
| Complication during last pregnancy | Yes | 78(15.3) | 54(10.6) | 5.1(3.57,7.22) *** | 3.4(1.94,6) *** |
| | No | 52(10.2) | 326(63.9) | 1 | 1 |
| Previous bad obstetrics history | Yes | 58(11.4) | 29(5.7) | 4(2.85,5.73)*** | 0.6(0.33,1.11) |
| | No | 72(14.2) | 351(68.8) | 1 | 1 |
| Type of pregnancy | Singleton | 113(22.2) | 299(58.6) | 1.5(0.90,2.51) | 0.83(0.44,1.58) |
| | Multiple | 17(3.3) | 81(15.9) | 1 | 1 |
| Onset of labor | Elective C/S | 21(4.1) | 59(11.6) | 1.34(0.83,2.16) | 2.3(0.95,5.39) |
| | Spontaneous | 91(17.8) | 284(55.7) | 1 | 1 |
| | Induced | 18(3.5) | 37(7.2) | 1.56(0.94,2.59) | 1.8(0.91,3.47) |
| Place of birth | Home | 57(11.2) | 35(6.9) | 4.1(2.77,5.99)*** | 1.2(0.68,2.00) |
| | Health center | 25(4.9) | 102(20) | 1.1(0.66, 1.75) | 1.1(0.61,2.01) |
| | Hospital | 48(9.4) | 243(47.6) | 1 | 1 |
| Mode of delivery | SVD | 93(18.2) | 299(58.6) | 1 | 1 |
| | CS | 37(7.2) | 81(15.9) | 1.66(1.13,2.45)* | 0.6(0.27,1.30) |
| Preeclampsia | Yes | 46(9) | 61(12) | 2.4(1.66,3.43)*** | 0.6(0.35,1.18) |
| | No | 84(16.5) | 319(62.5) | 1 | 1 |
| Diabetic mellitus | Yes | 10(2) | 7(1.4) | 2.28(1.19,4.35)* | 1.1(0.43,2.75) |
| | No | 120(23.5) | 373(73.1) | 1 | 1 |
| Gestational age | Extremely preterm | 14(2.7) | 25(4.9) | 1.73(0.94,3.18) | 1.5(0.71,3.32) |
| | very preterm | 35(6.9) | 44(8.6) | 2.32(1.48,3.64) *** | 1.4(0.76,2.44) |
| | moderate preterm | 39(7.6) | 99(19.4) | 1.5(0.97,2.33) | 1.2(0.76,2.06) |
| | late preterm | 42(8.2) | 212(41.6) | 1 | 1 |
| Weight for gestational age at birth | Small | 84(16.5) | 168(32.9) | 1.8(1.24,2.56) ** | 0.8(0.50,1.38) |
| | Appropriate | 46(9) | 212(41.6) | 1 | 1 |
| neonate weight(gm) | <1000 | 13(2.5) | 13(2.5) | 2(0.83, 4.87) | 0.9(0.29,2.67) |
| | 1000–1499 | 48(9.4) | 85(16.7) | 1.8(0.85,3.79) | 1.7(0.70,4.13) |
| | 1500–2499 | 61(12) | 246(48.2) | 1.16(0.56,2.44) | 1.7(0.76,3.92) |
| | ≥2500 | 8(1.6) | 36(7.1) | 1 | 1 |
| Neonate cry immediately at birth | Yes | 79(15.5) | 326(63.9) | 1 | 1 |
| | No | 51(10) | 54(10.6) | 2.36(1.65,3.36) *** | 0.6(0.35,1.16) |
| Bag and mask resuscitation at birth | Yes | 75(14.7) | 138(27.1) | 2.3(1.62,3.28) *** | 2.1(0.28,0.77) ** |
| | No | 55(10.8) | 242(47.4) | 1 | 1 |
| Peri-natal asphyxia diagnosed at birth | Yes | 46(9) | 37(7.2) | 2.74(1.91,3.93) *** | 1.1(0.57,2.15) |
| | No | 84(16.5) | 343(67.2) | 1 | 1 |

*(Continued)*

**Table 5.** (Continued)

| Variable | Category | Survival status | | CHR 95% CI | AHR 95% CI |
|---|---|---|---|---|---|
| | | Died (N, %) | Censored (N %) | | |
| Neonate diagnosed with RDS | Yes | 59(11.2) | 57(11.2) | 2.37(1.68,3.36) *** | 1.2(0.72,2.16) |
| | No | 71(13.9) | 323(63.3) | 1 | 1 |
| Hypoglycemia diagnosed at admission | Yes | 33(6.5) | 64(12.5) | 1.63(1.09,2.42) * | 1.5(0.88,2.42) |
| | No | 97(19) | 316(62) | 1 | 1 |
| Neonate diagnosed with sepsis | Yes | 96(18.8) | 203(39.8) | 1.75(1.19,2.60) ** | 0.9(0.55,1.44) |
| | No | 34(6.7) | 177(34.7) | 1 | 1 |
| (CPAP) | Yes | 52(10.2) | 91(17.8) | 1 | 1 |
| | No | 78(15.3) | 289(56.7) | 1.51(0.46, 0.94) * | 0.6(0.37,1.00) |
| KMC | Yes | 10(2) | 227(44.5) | 1 | 1 |
| | No | 120(23.5) | 150(30) | 12.64(6.62,24.13) *** | 9.3(4.36,19.9) *** |

(**Note**: AHR: adjusted hazard ratio; CHR: crude hazard ratio; CI: confidence interval

*p-value <0.05

**p-value <0.01

***p-value <0.001

characteristics including not receiving kangaroo mother care, lack of ante natal care follow-up, and intrapartum complications were identified to be predictors of time to death. As a result, high-risk newborns may receive priority care and appropriate attention.

Some limitations should also be considered. First, follow-up was limited to until discharge. Some post discharge events may have occurred after discharge. Second, due to the nature of the study design and incomplete records, certain major predictors of preterm mortality, such as mother demographics, nutritional condition, and institutional factors, were not addressed. Selection bias is possibly introduced during secondary data collection because patients with incomplete records were excluded so that the incidence of death may be underestimated.

The fact that data was collected retrospectively from secondary sources might also have an effect on the quality of the study.

## Conclusions

Despite admission to NICU for advanced care and follow up, mortality of preterm neonates was found to be high in the study settings. Addressing major intrapartum complications is required to improve survival of neonates admitted to NICU. Study on the appropriateness of care and delays in care is essential to fully understand the reasons behind high mortality.

## Supporting information

**S1 Dataset.**
(XLS)

## Acknowledgments

We would like to thank Haramaya University and Minster of Education support for data collection and we would like also to express our great appreciation to head of both hospitals and data collectors for their selfless provision of continuous support and facilitation during the data collection processes.

## Author Contributions

**Conceptualization:** Alo Edin Huka, Lemessa Oljira, Abera Kenay Tura, Angefa Ayele Tuluka.

**Data curation:** Alo Edin Huka, Adisu Birhanu Weldesenbet, Abdulmalik Abdela Bushra, Ibsa Abdusemed Ahmed, Angefa Ayele Tuluka.

**Formal analysis:** Alo Edin Huka, Adisu Birhanu Weldesenbet, Angefa Ayele Tuluka.

**Funding acquisition:** Alo Edin Huka.

**Investigation:** Alo Edin Huka.

**Methodology:** Alo Edin Huka, Lemessa Oljira, Adisu Birhanu Weldesenbet, Abera Kenay Tura, Angefa Ayele Tuluka.

**Project administration:** Alo Edin Huka.

**Resources:** Alo Edin Huka.

**Software:** Alo Edin Huka, Adisu Birhanu Weldesenbet, Abera Kenay Tura.

**Supervision:** Alo Edin Huka, Lemessa Oljira, Abera Kenay Tura.

**Validation:** Alo Edin Huka, Adisu Birhanu Weldesenbet, Abdulmalik Abdela Bushra, Ibsa Abdusemed Ahmed, Angefa Ayele Tuluka.

**Visualization:** Alo Edin Huka, Abdulmalik Abdela Bushra, Ibsa Abdusemed Ahmed.

**Writing – original draft:** Alo Edin Huka, Lemessa Oljira, Abera Kenay Tura, Angefa Ayele Tuluka.

**Writing – review & editing:** Alo Edin Huka, Lemessa Oljira, Adisu Birhanu Weldesenbet, Abdulmalik Abdela Bushra, Ibsa Abdusemed Ahmed, Abera Kenay Tura, Angefa Ayele Tuluka.

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
