## [Decision Letter · Decision Letter 0]

12 Dec 2022

PONE-D-22-26735Predictors of time to death among preterm neonates admitted to neonatal intensive care units at public hospitals in southern Ethiopia: A cohort studyPLOS ONE

Dear Dr. Huka,

Thank you for submitting your manuscript to PLOS ONE. After careful consideration, we feel that it has merit but does not fully meet PLOS ONE’s publication criteria as it currently stands. Therefore, we invite you to submit a revised version of the manuscript that addresses the points raised during the review process.

We look forward to receiving your revised manuscript.

Kind regards,

Ammal Mokhtar Metwally, Ph.D (MD)

Academic Editor

PLOS ONE

Journal Requirements:

“AEH received fund from Minster of Science and Higher Education, Ethiopia

Additional Editor Comments:

Please note that your manuscript was reviewed by 6 experts in the field. There is consensus agreement that the idea of the article is interesting. Meanwhile, some of the reviewers identified important problems in your submission and provided copious comments that should be considered .

Please note that further language improvement is indicated. Consider revising the spelling, grammar, diction, and syntax throughout the manuscript for increased clarity.

Reviewers' comments:

Reviewer's Responses to Questions

**Comments to the Author**

1. Is the manuscript technically sound, and do the data support the conclusions?

Reviewer #1: Partly

Reviewer #2: Yes

2. Has the statistical analysis been performed appropriately and rigorously? 

Reviewer #1: Yes

Reviewer #2: Yes

3. Have the authors made all data underlying the findings in their manuscript fully available?

Reviewer #1: Yes

Reviewer #2: Yes

4. Is the manuscript presented in an intelligible fashion and written in standard English?

Reviewer #1: Yes

Reviewer #2: Yes

5. Review Comments to the Author

Reviewer #1: Using a retrospective cohort study, the authors estimate the time to death and its predictors among preterm neonates admitted to NIC units at public hospitals in southern Ethiopia. The paper reports an overall incidence of neonatal mortality of 47.7 per 1000 neonate-days, with lack of antenatal care, pregnancy complication, resuscitation at birth, and not using skin-to-skin mother care method being the predictors identified.

First of all, this topic is of importance, given the number of newborn baby deaths, especially in sub-Saharan Africa. Such studies may aid in developing a roadmap towards achieving the SDGs by 2030.

With that in mind, this reviewer has the following to remark:

1. The text appears to be in need of some editorial attention to improve readability.

2. It is recommended that the authors note the criteria used with respect to the inclusion of variables in cox regression analyses.

3. Did the authors calculate the maximum number of variables to include in the study?

4. I wonder if the authors thought of categorizing their variables into fixed versus time-dependent covariates.

5. In the results section, it is stated that “in the bi-variable cox regression analysis, maternal residence, antenatal care follow up, complication during last pregnancy, previous bad obstetrics history, place of birth, mode of delivery, preeclampsia, diabetic mellitus, gestational age, weight for gestational age, neonate cried immediately at birth, bag and mask resuscitation at birth, perinatal asphyxia at birth, neonate diagnosed with respiratory distress, hypoglycemia, neonatal sepsis, phototherapy, nasal continuous positive airway pressure, not receiving kangaroo mother care and radiant warmer were predictors of mortality at p<0.25.”

The abstract mentions that “Cox-Proportional Hazards model was used to identify significant predictors of time to death at p<0.05.”

To avoid potential confusion, it is recommended that the authors share briefly the rationale behind calling the above factors “predictors of mortality” when statistical significance wasn’t observed at the 5% level.

6. Did the authors assess the presence of interaction or effect modification?

7. A last point involves the limitations mentioned at the end of the discussion section. They are not sufficient. This paper would be stronger if it includes other obvious limitations.

I hope this review is helpful and wish the authors the very best with their research!

Reviewer #2: Title: "Predictors of time to death among preterm neonates admitted to neonatal intensive care units at public hospitals in southern Ethiopia: A cohort study."

Summary comments: Generally well written manuscript. The subject of the paper, neonatal mortality is an important subject of public health importance.

Few comments below mainly related to typos should be addressed by the authors

Introduction

Line 46: Authors wrote “Preterm birth, defined as a live birth before 37 completed weeks of pregnancy(1) and remains one of the common causes of hospitalization……..”

They may either capture the sentence as “Preterm birth, defined as a live birth before 37 completed weeks of pregnancy(1) remains one of the common causes of hospitalization……..

”

Or “Preterm birth is defined as a live birth before 37 completed weeks of pregnancy(1) and remains one of the common causes of hospitalization ………

”

Line 49-51. The authors should review the tenses used. For instance, they state “Globally, the sub-Saharan Africa had the highest newborn mortality rate 27 deaths per 1,000 live births, followed by Central and Southern Asia (25 deaths per 1,000 live births) (3).

”

They should clarify whether they mean, Globally, sub-saharan Africa has… or they meant ‘’had’’ in which case it will be good to know what the current NMR is sub-Saharan Africa is.

Line 56-57: The sentence below is incomplete, please rephrase into a complete sentence: “Despite the national goal of reducing the NMR from 57 28 to 11 per 1000 by 2020 as part of the national newborn and child survival strategy (2015-2020) 58 (6).”

Line 54-58. Question: Why do we have differences in the national Health Sector Transformation Plan (HSTP-I) target of reducing NMR to 10 deaths per 1,000 live births by 2020 and the national goal of reducing the NMR from 28 to 11 per 1000 by 2020. Are they different agencies?

Line 62: “Ethiopia's neonatal mortality rate unacceptably high”

It should rather read “Ethiopia's neonatal mortality rate is unacceptably high”

Methods

Lines 76 and 80, please remove duplication of sentence as captured below:

Line 76: neonatology ward accommodate 16beds in including 8 total beds from NICU ward.

Line 80: there is 16 beds in neonatology ward including 8 total beds from NICU ward.

Comment: Repetition of sentence

Line 97 to 99:

“The checklist for this study was adapted from the similar literatures, neonates’ medical card and

99 logbook and modified after reviewing of other related literatures.”

This may read better as “The checklist for this study was adapted from neonates’ medical card and logbook with modifications based on literature review.” Avoid use of the term “literatures”. Literature can be used as plural.

Line 99: “A checklist consisted of the…”

Please use “The checklist consisted of the…..”

Line 97 to 104: In summary, the section on data collection can be better organized.

See suggestion below based on what the authors been wrote:

“The data were extracted from each individual neonate’s medical card and logbook using a structured checklist. The checklist consisted of information on maternal and neonatal socio demographic factors, neonatal related factors, maternal medical and obstetric-related factors.

Since the data was collected from neonates’ medical card, the data collectors were trained on how to extract appropriate data and there was close daily monitoring and supervision at the data collection site.”

Results: Line 153: “Four hundred and five neonates (79%) were cried immediately”

Comment; Please remove “were”

Discussion:

Line 276-277: “This study revealed that 9-fold increase in hazard to mortality among preterm neonates who had not received KMC compared to those who were received KMC during hospital stay.”

Rephrase: “This study revealed 9-fold increase in hazard to mortality among preterm neonates who had not received KMC compared to those who received KMC during hospital stay.”

The manuscript will benefit from a conclusion paragraph that summarizes the findings and way forward. The conclusion at the abstract section is appropriate and can be used or adapted for this section.

See conclusion of abstract below: “Conclusion: Despite admission to NICU for advanced care and follow up, mortality of preterm neonates was found to be high in the study settings. Addressing major intrapartum complications is required to improve survival of neonates admitted to NICU.”

6. PLOS authors have the option to publish the peer review history of their article (what does this mean?). If published, this will include your full peer review and any attached files.

Reviewer #1: No

Reviewer #2: **Yes: **Dr. Richard Akuffo

---

## [Author Response · Author response to Decision Letter 0]

25 Jan 2023

Dear Ammal Mokhtar Metwally, Ph.D (MD),

 Academic Editor

 PLOS ONE

Dear Editor, this is regarding the manuscript PONE-D-22-26735 entitled as “Predictors of time to death among preterm neonates admitted to neonatal intensive care units at public hospitals in southern Ethiopia: A cohort study.” submitted to PLOS ONE. Thank you for the opportunity to respond to the constructive reviews provided to our submission to PLOS One. We are grateful that reviewers 1 partly and 2 agree that this is a technically sound piece of research and hope we have satisfied both reviewers concerns about the rigor of the statistical assessment. We appreciate that both reviewers agree we have been transparent with provision of the data on which our conclusions are based and have presented the results in intelligible language. We have addressed all reviewer comments in the revised manuscript now respectfully submitted as both marked (Revised Manuscript with Track Changes) and unmarked (Manuscript) versions. 

We have made every effort to comply with the PLOS One Style requirements in accord with journal templates. Before addressing the specific reviewer comments point-by-point below, we would like to respond to your editorial request as follow:-

Journal Requirements:

Response: Dear academic editor, Thank you so much for your suggestion. We have carefully considered and it has been done in the revised version.

“AEH received fund from Minster of Education, Ethiopia 

Response: Dear academic editor, Thank you so much for your suggestion. We apologize for first funding statement that was submitted as “Minster of Science and Higher Education “Funded this study and again We apologize for any inconvenience this has caused.

 However, as per funding statement, we would like to confirm that the following statement is accurate. 

“AEH received fund from Haramaya University, Ethiopia.

Response: We have uploaded our supplementary information, including the full data set, as Supporting Information files (Supporting information, Line 326, S3 Data set). And under Data Availability statement we have modified as “All necessary information were included with the manuscript and its Supporting Information files”

Response: Thank you very much for your suggestion. We have removed ethical statement from other part of the manuscript except in Methods section.

Additional Editor Comments:

Please note that your manuscript was reviewed by 6 experts in the field. There is consensus agreement that the idea of the article is interesting. Meanwhile, some of the reviewers identified important problems in your submission and provided copious comments that should be considered 

Please note that further language improvement is indicated. Consider revising the spelling, grammar, diction, and syntax throughout the manuscript for increased clarity.

Response: Dear academic editor, we are thankful for your important comment and we have tried to edit the grammatical flaws throughout the manuscript in its revised version. We have edited the spelling, grammatical errors, diction, and syntax throughout the manuscript. Now we believe that the revised version is clean and clear enough to the readers.

 5. Review Comments to the Author

Reviewer #1: Using a retrospective cohort study, the authors estimate the time to death and its predictors among preterm neonates admitted to NIC units at public hospitals in southern Ethiopia. The paper reports an overall incidence of neonatal mortality of 47.7 per 1000 neonate-days, with lack of antenatal care, pregnancy complication, resuscitation at birth, and not using skin-to-skin mother care method being the predictors identified.

First of all, this topic is of importance, given the number of newborn baby deaths, especially in sub-Saharan Africa. Such studies may aid in developing a roadmap towards achieving the SDGs by 2030.

With that in mind, this reviewer has the following to remark:

1. The text appears to be in need of some editorial attention to improve readability.

Response: Dear reviewer, we are thankful for your important comment and we have tried to edit the grammatical flaws throughout the manuscript in its revised version. We have edited the spelling, grammatical errors, incomplete and poorly structured sentences throughout the manuscript. Now we believe that the revised version is clean and clear enough to the readers. 

2. It is recommended that the authors note the criteria used with respect to the inclusion of variables in cox regression analyses.

Response: Thank you Dear reviewer, the authors used following criteria for inclusion of variables in cox regression analyses.

 First, in Bivariable analysis, variables with p value < 0.25 were considered as candidate for multivariable model and included in the final Cox-regression analysis and We set a lower threshold for the p value (< 0.25) and giving priority to clinical reasoning in selecting variables, along with statistical significance, perhaps we would have allowed more important variables to be entered into the model. Any statistical test is considered significant at p-value < 0.05 (Data processing and analysis, Line 113-116)

3. Did the authors calculate the maximum number of variables to include in the study?

Response: Dear Reviewer, thank you very much for your time and consideration. We didn’t calculate the maximum number of the variables initially, but, at design stage the authors have reviewed all relevant literature and clinical experiences based on this, we have included all potential independent variables. Similarly, at the analysis level first we have conducted crude analysis to identify candidate variable using cut off p value < 0.25 for multivariable analysis model. Accordingly, we have conducted adjusted analysis to determine significant predictors of time to death at p-value < 0.05.

4. I wonder if the authors thought of categorizing their variables into fixed versus time-dependent covariates.

Response: Dear Reviewer, we found this comment is relevant. It is good to categorize the variables into fixed and time-dependent covariates in the longitudinal data analysis. But, in this study all time dependent covariates like age, weight and the like were assessed at the base line. 

5. In the results section, it is stated that “in the bi-variable cox regression analysis, maternal residence, antenatal care follow up, complication during last pregnancy, previous bad obstetrics history, place of birth, mode of delivery, preeclampsia, diabetic mellitus, gestational age, weight for gestational age, neonate cried immediately at birth, bag and mask resuscitation at birth, perinatal asphyxia at birth, neonate diagnosed with respiratory distress, hypoglycemia, neonatal sepsis, phototherapy, nasal continuous positive airway pressure, not receiving kangaroo mother care and radiant warmer were predictors of mortality at p<0.25.”

The abstract mentions that “Cox-Proportional Hazards model was used to identify significant predictors of time to death at p<0.05.”

To avoid potential confusion, it is recommended that the authors share briefly the rationale behind calling the above factors “predictors of mortality” when statistical significance wasn’t observed at the 5% level.

Response: Thank you Dear Reviewer, just this is to mean that these variables were significant predictors of time to death at p<0.05 in bivariable cox regression analysis. We have modified the sentence to “… at p<0.05..”) in the revised manuscript.

6. Did the authors assess the presence of interaction or effect modification?

Response: Thank you Dear Reviewer, We appreciate your suggestion of assessing effect modification. Yes, it was assessed. First, we have conducted a crude analysis of the relationship between the exposure variables and the event of interest. Next, we conducted a stratified analysis that separates the crude data according to levels of the effect modifiers. Effect measure modification was evaluated by examining the stratum-specific estimates. Therefore, based on this we did not identify effect modifiers in our findings.

7. A last point involves the limitations mentioned at the end of the discussion section. They are not sufficient. This paper would be stronger if it includes other obvious limitations.

Response: We thank the reviewer for the comment on limitations. We have recognized these things in the revised manuscript, on study limitations with the statement (Line 294-297) “Selection bias is possibly introduced during secondary data collection because patients with incomplete records were excluded so that the incidence of death may be underestimated….” and we have added the limitation due to “…data was collected retrospectively from secondary sources might also have an effect on the quality of the study...” 

Reviewer #2: Title: "Predictors of time to death among preterm neonates admitted to neonatal intensive care units at public hospitals in southern Ethiopia: A cohort study."

Summary comments: Generally well written manuscript. The subject of the paper, neonatal mortality is an important subject of public health importance.

Few comments below mainly related to typos should be addressed by the authors

Introduction

Line 46: Authors wrote “Preterm birth, defined as a live birth before 37 completed weeks of pregnancy(1) and remains one of the common causes of hospitalization……..”

They may either capture the sentence as “Preterm birth, defined as a live birth before 37 completed weeks of pregnancy(1) remains one of the common causes of hospitalization……..

” Or “Preterm birth is defined as a live birth before 37 completed weeks of pregnancy(1) and remains one of the common causes of hospitalization ………”

Response: Dear Reviewer, thank you very much for your time and consideration in editing and reviewing our manuscript. We have carefully read your comments and corrected inline of reviewer’s comments and suggestions. All comments raised were edited and incorporated in the main manuscript. The changes were highlighted with yellow color in the manuscript.

Line 49-51. The authors should review the tenses used. For instance, they state “Globally, the sub-Saharan Africa had the highest newborn mortality rate 27 deaths per 1,000 live births, followed by Central and Southern Asia (25 deaths per 1,000 live births) (3).”

They should clarify whether they mean, Globally, sub-saharan Africa has… or they meant ‘’had’’ in which case it will be good to know what the current NMR is sub-Saharan Africa is.

Response: We appreciate that the reviewer asked for more accurate wording regarding had. We have substituted “had” by “has” which is appropriate verb to state the current status of NMR in sub-Saharan Africa.

Line 56-57: The sentence below is incomplete, please rephrase into a complete sentence: “Despite the national goal of reducing the NMR from 57 28 to 11 per 1000 by 2020 as part of the national newborn and child survival strategy (2015-2020) 58 (6).”

Response: Thank you Dear reviewer, we have paraphrased this sentence to”… Neonatal mortality in Ethiopia is still high despite the country's goal of reducing it from 28 to 11 per 1000 by 2020 as part of the national newborn and child survival strategy (2015-2020) …

Line 54-58. Question: Why do we have differences in the national Health Sector Transformation Plan (HSTP-I) target of reducing NMR to 10 deaths per 1,000 live births by 2020 and the national goal of reducing the NMR from 28 to 11 per 1000 by 2020. Are they different agencies?

Response: Dear reviewer, thank you very much for your time and consideration. They are all part of the Ethiopian Ministry of Health sectors, but each has its own set of goals for reducing newborn mortality in accordance with other global targets. These sectors are:

First is “Ministry of Health. Health Sector Transformation Plan I (HSTP-I) 2015/16- 2019/20. MOH, 2015” with target of reducing NMR to 10 deaths per 1,000 live births by 2020.

Second is “Ministry of Health. National Newborn and Child Survival Strategy 2015/16-2019/20. MOH, 2015” with goal of reducing the NMR from 28 to 11 per 1000 by 2020. 

However, in both contests, Ethiopia is far behind achieving these national and international targets.

Line 62: “Ethiopia's neonatal mortality rate unacceptably high”

It should rather read “Ethiopia's neonatal mortality rate is unacceptably high”

Response: Thank you Dear reviewer, we have paraphrased this sentence to “Ethiopia's neonatal mortality rate is unacceptably high”.

Methods

Lines 76 and 80, please remove duplication of sentence as captured below:

Line 76: neonatology ward accommodate 16beds in including 8 total beds from NICU ward.

Line 80: there is 16 beds in neonatology ward including 8 total beds from NICU ward.

Comment: Repetition of sentence

Response: Thank you dear, we have removed duplicated sentence and it is corrected.

Line 97 to 99:

“The checklist for this study was adapted from the similar literatures, neonates’ medical card and

99 logbook and modified after reviewing of other related literatures.”

This may read better as “The checklist for this study was adapted from neonates’ medical card and logbook with modifications based on literature review.” Avoid use of the term “literatures”. Literature can be used as plural.

Line 99: “A checklist consisted of the…”

Please use “The checklist consisted of the…..”

Response: Thank you so much for your suggestion. We have accepted your comment in the revised manuscript.

Line 97 to 104: In summary, the section on data collection can be better organized.

See suggestion below based on what the authors been wrote:

“The data were extracted from each individual neonate’s medical card and logbook using a structured checklist. The checklist consisted of information on maternal and neonatal socio demographic factors, neonatal related factors, maternal medical and obstetric-related factors.

Since the data was collected from neonates’ medical card, the data collectors were trained on how to extract appropriate data and there was close daily monitoring and supervision at the data collection site.”

Response: Dear reviewer, we found this comment is relevant and we have corrected all as per suggestion.

Results: Line 153: “Four hundred and five neonates (79%) were cried immediately”

Comment; Please remove “were”

Response: Thank you for this correction and the comment is accepted.

Discussion:

Line 276-277: “This study revealed that 9-fold increase in hazard to mortality among preterm neonates who had not received KMC compared to those who were received KMC during hospital stay.”

Rephrase: “This study revealed 9-fold increase in hazard to mortality among preterm neonates who had not received KMC compared to those who received KMC during hospital stay.”

Response: Dear reviewer, we found this comment is relevant and we have corrected all as per comment. We have incorporated all editorial and reviewer comments.

The manuscript will benefit from a conclusion paragraph that summarizes the findings and way forward. The conclusion at the abstract section is appropriate and can be used or adapted for this section.

See conclusion of abstract below: “Conclusion: Despite admission to NICU for advanced care and follow up, mortality of preterm neonates was found to be high in the study settings. Addressing major intrapartum complications is required to improve survival of neonates admitted to NICU.”

Response: Thank you Dear reviewer, we have accepted your comment in the revised manuscript.

We thank again all editors and reviewers for their helpful guidance, and critiques that have strengthened this research report

---

## [Decision Letter · Decision Letter 1]

3 Mar 2023

Predictors of time to death among preterm neonates admitted to neonatal intensive care units at public hospitals in southern Ethiopia: A cohort study

PONE-D-22-26735R1

Dear Dr. Huka,

We’re pleased to inform you that your manuscript has been judged scientifically suitable for publication and will be formally accepted for publication once it meets all outstanding technical requirements.

Kind regards,

Ammal Mokhtar Metwally, Ph.D (MD)

Academic Editor

PLOS ONE

Additional Editor Comments (optional):

Reviewers' comments:

Reviewer's Responses to Questions

**Comments to the Author**

1. If the authors have adequately addressed your comments raised in a previous round of review and you feel that this manuscript is now acceptable for publication, you may indicate that here to bypass the “Comments to the Author” section, enter your conflict of interest statement in the “Confidential to Editor” section, and submit your "Accept" recommendation.

Reviewer #1: All comments have been addressed

Reviewer #2: All comments have been addressed

2. Is the manuscript technically sound, and do the data support the conclusions?

Reviewer #1: (No Response)

Reviewer #2: Yes

3. Has the statistical analysis been performed appropriately and rigorously? 

Reviewer #1: (No Response)

Reviewer #2: Yes

4. Have the authors made all data underlying the findings in their manuscript fully available?

Reviewer #1: (No Response)

Reviewer #2: Yes

5. Is the manuscript presented in an intelligible fashion and written in standard English?

Reviewer #1: (No Response)

Reviewer #2: Yes

6. Review Comments to the Author

Reviewer #1: Based on feedback from the editor and reviewers, the article has been revised and is now more concise.

Once again, I wish the authors the best of luck with their research!

Reviewer #2: The authors have adequately addressed all previous review comments made. The manuscripts reads better now.

7. PLOS authors have the option to publish the peer review history of their article (what does this mean?). If published, this will include your full peer review and any attached files.

Reviewer #1: No

Reviewer #2: **Yes: **Dr. Richard Adjei Akuffo

---

## [Editor Report · Acceptance letter]

4 Oct 2023

PONE-D-22-26735R1 

Predictors of time to death among preterm neonates admitted to neonatal intensive care units at public hospitals in southern Ethiopia: A cohort study 

Dear Dr. Huka:

I'm pleased to inform you that your manuscript has been deemed suitable for publication in PLOS ONE. Congratulations! Your manuscript is now with our production department. 

Kind regards, 

on behalf of

Professor Ammal Mokhtar Metwally 

Academic Editor

PLOS ONE